# Impact of Gait and Diameter during Circular Exercise on Front Hoof Area, Vertical Force, and Pressure in Mature Horses

**DOI:** 10.3390/ani11123581

**Published:** 2021-12-17

**Authors:** Alyssa A. Logan, Brian D. Nielsen, Cara I. Robison, David B. Hallock, Jane M. Manfredi, Kristina M. Hiney, Daniel D. Buskirk, John M. Popovich

**Affiliations:** 1Department of Animal Science, Michigan State University, 474 S. Shaw Ln., East Lansing, MI 48824, USA; bdn@msu.edu (B.D.N.); oconn107@msu.edu (C.I.R.); buskirk@msu.edu (D.D.B.); 23R Forge and Farriery, Dansville, MI 48819, USA; 3rforge@gmail.com; 3Department of Pathobiology and Diagnostic Investigation, Michigan State University, 784 Wilson, Rd., East Lansing, MI 48824, USA; manfred1@msu.edu; 4Department of Animal and Food Sciences, Oklahoma State University, 201J Animal Sciences, Stillwater, OK 74074, USA; khiney@okstate.edu; 5Department of Osteopathic Surgical Specialties, Michigan State University, 909 Fee Rd., B405, East Lansing, MI 48824, USA; popovi16@msu.edu

**Keywords:** lunge, round pen, bone, joint, canter, Tekscan

## Abstract

**Simple Summary:**

Circular exercise is used frequently to exercise, train, and evaluate horses both under saddle and with lunging. However, little is known of the impacts this type of repetitive exercise has on the front limbs of horses. Nine mature horses wore Tekscan^TM^ Hoof Sensors on their front hooves to determine if changing the circle size and gait at which the horse is traveling impacts the area, vertical force, or pressure output. Sensor data were collected while horses travelled in a straight line at the walk and trot and in small and large counterclockwise circles at the walk, trot, and canter. Gait was found to be a driving factor for differences in outputs, with mean area, mean vertical force, and mean pressure being greater at the walk in a straight line, and the area being greater at the canter when circling. When traveling in a counterclockwise circle, the mean area of the outside front leg was highest at the canter. This study shows gait is an important factor when evaluating exercise in a circle or straight line. Horse owners may choose to perform circular exercise at slower gaits or minimize unnecessary circular exercise to decrease differences between limbs and potentially reduce injury.

**Abstract:**

Circular exercise can be used at varying gaits and diameters to exercise horses, with repeated use anecdotally relating to increased lameness. This work sought to characterize mean area, mean vertical force, and mean pressure of the front hooves while exercising in a straight line at the walk and trot, and small (10-m diameter) and large circles (15-m diameter) at the walk, trot, and canter. Nine mature horses wore Tekscan^TM^ Hoof Sensors on their forelimbs adhered with a glue-on shoe. Statistical analysis was performed in SAS 9.4 with fixed effects of leg, gait, and exercise type (PROC GLIMMIX) and *p* < 0.05 as significant. For all exercise types, the walk had greater mean pressure than the trot (*p* < 0.01). At the walk, the straight line had greater mean area loaded than the large circle (*p* = 0.01), and both circle sizes had lower mean vertical force than the straight line (*p* = 0.003). During circular exercise at the canter, the outside front limb had greater mean area loaded than at the walk and trot (*p* = 0.001). This study found that gait is an important factor when evaluating circular exercise and should be considered when exercising horses to prevent injury.

## 1. Introduction

The use of circular exercise is frequent in equine training, both under-saddle and in-hand via lunging, and anecdotally has the potential to contribute to lameness. During early training, horses are often exercised in a circular manner through lunging or in a round pen. Some riding disciplines, such as dressage, reining, and barrel racing, use circular exercise during training and competition throughout a horse’s career. Often, the circles performed in these disciplines are on a small radius with high speed and are utilized frequently within a training session. Thoroughbred racehorses also experience circular forces as they lean into a bend at high speeds [1,2]. Lunging with and without lunging aids, and the use of mechanical horse walkers is found in many rehabilitation protocols [3,4]. When surveyed, 50% of Thoroughbred trainers in Victoria, Australia indicated the use of a mechanical walker as an alternative exercise method to track work [5]. In lameness evaluations, a higher proportion of lameness can be found while utilizing lunging on hard or soft surfaces compared to straight lines [6]. While circular exercise is commonly used, many in the industry are unaware of the potential negative impacts it can have on joint health. While being exercised in a circle, horses will lean into the continuous turns up to 20 degrees to maintain balance. As speed and curve increase, the lean angle will also increase [1,7,8,9,10]. Greater tilt on a flat curved surface has also been found compared to a banked curve surface [10]. Due to the possibility of a reduced loaded area, uneven vertical forces may be placed on joints and bones of the limbs during circular exercise.

Quadrupeds, such as horses, may be at an advantage during curve running, as they can redistribute weight to multiple stance legs within a stride [1]. It has been found that, while cantering in a 10-m circle, horses will have greater peak ground reaction force on the outside forelimb compared to the inside forelimb. This difference in vertical forces between limbs was not evident at slower speeds and may be due to the presence of a lead and non-lead forelimb and hindlimb during the canter. In both humans and horses, the outside limb while sprinting through a curve is known to generate more vertical and lateral force than the inside limb [1]. When speed is held constant between a straight line and a curve, stride duration is seen to increase when horses travel around a curve compared to a straight line. With training, this increase in stride duration around a curve is seen to decrease, potentially through familiarity via training or neuromuscular adaptation [2].

There are few equine studies evaluating circular exercise; however, human exercise studies evaluating running on a curve are abundant. It has been found that in humans, when running around an unbanked curve, the inside leg has a lower peak ground reaction force than the outside leg. Peak ground reaction forces of both legs and speed are also lower when sprinting around a curve compared to in a straight line [11]. Not only does the presence of a curve impact runners, so does the sharpness of a curve. Running on a sharply curved track (5-m radius) led to greater torsion on the inside tibia compared to running on a gently curved track (15-m) or a straight line [12]. Both the speed and the radius of a circle will impact the gait asymmetry of horses circling at the trot [9]. A retrospective study of risk factors for jockeys in Japanese Thoroughbred racing found smaller tracks to have a greater risk of injury [13].

In racing Quarter Horses, who race on straight portions of tracks, the right forelimb is most commonly involved in catastrophic musculoskeletal injuries (CMIs), with the left forelimb following (57% and 24% of CMIs, respectively). This may be due to the preference of the right lead in racing Quarter Horses [14]. However, the presence of motor laterality within the equine population is questionable and may not exist outside of horses in race training, as it was not found in Quarter Horses trained for cutting [15]. Thoroughbreds typically race counterclockwise on an oval track in North America, with the left front leg as the leading leg while traveling around a curve in the track. A study of Midwestern U.S. racetracks found that the left forelimb was the most common site for injuries (56% of CMIs) in Thoroughbreds, while the right forelimb was the most common (60% of CMIs) in Quarter Horses [16]. As a result of greater impulse on the left forelimb while galloping around the turn of a race track, Thoroughbred racehorses are at greater risk for injury to the left forelimb while traveling around a turn [17].

It has been hypothesized that horses handle traveling in a circle similar to the adjustments they make in response to lameness on a straight line; horses counteract uncomfortable limb loading with asymmetrical movement, which redistributes limb loading [18]. Horses will lean into the circle in which they are traveling, with the lean increasing with speed and smaller radii [9]. The level of training can also impact lean angle, as a horse acclimated to tracking in a circle can travel more upright than a horse that is not acclimated to tracking in a circle or bend. This has especially been noted in dressage, where older, trained horses are able to travel while engaging their neck, back, and hindlimb musculature, where a younger horse may not be able to do so through a circle. Body lean may also be increased when a lame forelimb is on the inside of the circle [8,10,19]. When circling at the trot, a two-beat diagonal gait, horses have decreased loading when the inside front leg and outside hind leg are in a stance, compared to a push-off pattern for the outer limbs [20,21].

Osteoarthritis (OA) and joint injuries have been reported as a leading cause of lost training days and horse wastage. There may be a connection between OA and circular exercise, but interactions between circular exercise and joint damage have not been explored. With up to 60% of lameness being related to OA, this gap in research greatly affects the equine community [14,22]. Within the United States, Quarter Horses and Thoroughbreds have been identified as making up half the population affected by OA [23]. Osteoarthritis can occur secondary to excessive loads on normal cartilage or normal loads on abnormal cartilage [24,25], with both mechanisms possibly exacerbated by circular exercise. It has been noted that horses are able to perform adaptations to limb position while exercising on a circle via abduction (pushing the limbs away from the midline of the body) and adduction (bringing the limbs towards the midline), but these adaptations performed over long periods of time and at faster gaits may lead to greater risk of injury to the distal limb [10,25].

Utilizing the Tekscan^TM^ Hoof System (Tekscan, Inc., Boston, MA, USA), the aim of this study was to categorize the outputs (i.e., area, vertical force and pressure) for the front hooves of horses during counterclockwise circular exercise, and to demonstrate that these outputs vary depending on circle diameter and gait. It was hypothesized that faster gaits and a decreased circle diameter would lead to greater disparity in the mean area, vertical force, and pressure of the front limbs, with the outer limbs having a smaller mean loaded area with greater mean vertical force and pressure.

## 2. Materials and Methods

This research was approved by the Michigan State University (MSU) Institutional Animal Care and Use Committee (PROTO201800148).

### 2.1. Horses

A total of nine mature horses participated in this study (14 ± 2 years). Arabian horses were obtained from the MSU Horse Teaching and Research center (*n* = 4: Two mares and two geldings) and stock horses were obtained from a local training operation (*n* = 5: Two mares and three geldings). One week prior to beginning exercise, horses were evaluated by two board-certified veterinarians (large animal surgery and one also boarded in equine sports medicine and rehabilitation) with a Lameness Locator^®®^ and subjective lameness evaluation to be sound (American Association of Equine Practitioners lameness grade of <2 on each front leg). Horses were transported to the MSU Pavilion South Arena for one day for exercise analysis. During the day, while not being exercised, horses were given ad libitum access to water and hay and kept in individual stalls.

### 2.2. Hoof Preparation and Sensors

Hoof and sensor preparations were performed in a method previously utilized when using the Tekscan Hoof System^TM^ with a glue-on shoe [26]. Horses were trimmed by a certified farrier (Certified Journey Farrier, Advanced Skill Farrier, Associate of the Worshipful Company of Farriers) within a week before exercise. Horses were trimmed for medial-lateral balance according to the long axis of the limb observed with the hoof picked up. Excess hoof was removed as needed to achieve a flat plane to place sensors, and shoe placement was guided by the center of rotation (COR) [27]. Horse hooves were measured for width and length to determine an accurate glue-on shoe (SoundHorse Technology, Unionville, PA, USA) size for each horse (Table 1). An exact fit for each front hoof and shoe was desired so that the loaded sensor area represented the loaded area of the hoof. Tekscan hoof sensors were trimmed to the size of the front hooves of each of the horses after initial hoof trimming. The trimmed sensors were sealed in two layers of liquid rubber (FlexSeal, Weston, FL, USA) to protect the sensors from moisture and sand exposure. The liquid rubber sealing was allowed to dry for 24–48 h before sensors were placed on horses.

The day before horses exercised, horses were weighed for calibration. Two scales of equal height were used. Horses stood with their hind legs on one scale and their front legs on another and were encouraged to stand square with their weight equally distributed. The weight of the front half of the horse was recorded on one scale for sensor calibration (Tru-Test Multipurpose MP600 Load Bars). This weight was divided in half, to represent the left limb and the right limb (Table 1).

On the day of exercise, the sealed and trimmed sensors were attached to the front hooves of each horse with a glue-on shoe and animal-safe epoxy. Horses were shod with a ratio of 60% in front of COR and 40% behind [27]. After the initial mixing, the two-part epoxy used to adhere the shoe to the hoof wall was dried for approximately 30 min to be cured for exercise. Before beginning exercise, the sensors were calibrated with the previously recorded weight of the forelimbs. Each horse walked the length of the indoor arena to pre-load the sensors. Afterwards, they were brought to a flat spot in the middle of the arena and encouraged to stand squarely with their weight evenly distributed. Using F-Scan research software (Tekscan^TM^), the previously determined weight of the front limbs (Table 1) was inputted with the step calibration function, and a calibration file was saved for the left and right forelimbs for each horse.

### 2.3. Exercise

Previous research with the Tekscan Hoof^TM^ system has found that when used with a glue-on shoe on the front hooves, the sensors are reliable within a session of exercise [26]. Bearing this in mind, the current study was designed that each horse would complete all their exercise within one session, and each set of sensors would only be used once. Straight-line exercise was performed first for each horse, so that if a sensor was damaged during circular exercise, then all horses would have straight-line exercise recordings. The space in which straight-line exercise was recorded was 25 m in length. Multiple recordings have previously been suggested for exercise protocols utilizing sensors such as this to gain a mean of the desired outputs [28,29,30,31]. Each horse was recorded three times at both the walk and trot traveling in a straight line the length of the indoor arena. The canter was not recorded in a straight line, as the gait is not able to be safely and consistently attained when horses are led in-hand. Each recording included at least 10 steps of the horse performing the specified gait consistently with no break. All straight-line exercise was performed by the same handler.

After the straight-line exercise, each horse was led to the portable round pen in the middle of the indoor arena. Order of size for the circular exercise (small first or large first) was randomly assigned for each horse. The size of the small circle was 10 m in diameter, while the size of the large circle was 15 m. Circle diameter was adjusted by adding or removing portable round pen panels with the perimeter for each circle size marked in the sand of the arena so that both circles were set up the same each time. For each horse, three recordings of at least 10 strides at the walk, trot, and canter were taken for both the large circle and the small circle. All circular exercise was performed in a counterclockwise direction, with the left forelimb on the inside and the right forelimb on the outside. Only one direction was evaluated so that one limb could be consistently denoted as the outside limb and the other as the inside limb. Counterclockwise was chosen as this is the direction of travel for racing horses and is typically the first direction of travel for horses competing in judged classes on the rail. Horses were encouraged to maintain gait speed with a human handler either verbally or visually encouraging them to gain speed or slow down. The speed of the walk, trot, or canter was not controlled between animals, as each individual animal has a speed for each gait at which they are able to move comfortably and maintain their gait consistently through recordings. Other gait analysis studies have preferred to allow animals to travel at their natural speed within a gait during testing [6,20,22,29]. If there were errors such as an incorrect lead or break of gait, the recording in process was stopped and re-recorded. Following exercise, the glue-on shoes and sensors were removed from each horse.

### 2.4. Data Analysis

Sensor data were recorded at a sampling rate of 112 frames/second for all conditions. Recorded data were then analyzed with Tekscan F-Scan Software (version 6.85). F-Scan Software collected vertical force and area outputs, and calculated pressure from the coordinating vertical force and area for a frame. The first and last steps were removed from each recording dataset to ensure that no transitional steps between gaits were included in the dataset. Each recording dataset still had at least 10 steps with the first and last step removed. If individual sensor cells, sensels, were loaded during a suspension phase for a hoof, these erroneous sensels were manually removed. The mean vertical force, area, and pressure for each step were exported from F-Scan Software in an ASCII file.

### 2.5. Statistical Analysis

Data were exported from the F-Scan Software (version 6.85) and imported into SAS (version 9.4) for statistical analysis and were evaluated for normality via residual plots. To evaluate the impacts that gait and circle size have on kinematic outputs, two datasets were created. One dataset removed the gait “canter”, so that at both the walk and trot outputs could be compared for the straight line and both circle sizes. A second dataset removed the exercise type “straight”, so that at the walk, trot, and canter, the two circle sizes could be compared. These two separate datasets were necessary, as a canter on a straight line was not able to be safely included in the data collection for this study. The main effects of gait, exercise type, and leg were evaluated in PROC GLIMMIX with Tukey adjustment. Interactions of “gait and exercise type”, “gait and leg”, “exercise type and leg”, and “gait, exercise type, and leg” were also evaluated. Horse, and all interactions including horse, was included as a random effect. Significance was set at *p* < 0.05. Means are reported at means ± standard error of the mean (SEM).

## 3. Results

### 3.1. Dataset without Canter

#### 3.1.1. Loaded Hoof Area

Gait (*p* = 0.03) and exercise type (*p* = 0.03) were significant main effects for area (Table 1), but leg was not (*p* = 0.44). The walk had a greater mean area than the trot by 12%. Gait and exercise type constituted a significant interaction (*p* = 0.005), but “gait and leg” as well as “exercise type and leg” were non-significant interactions (*p* = 0.10 and 0.71, respectively). “Gait, exercise type, and leg” was not a significant interaction (*p* = 0.48). At the walk and trot, there were no significant between-leg (left vs. right) differences in the mean area (*p* = 0.33 and *p* = 0.58, respectively)

At the walk, the area was different between exercise types (*p* = 0.01, Table 2), but not at the trot. While trotting in a straight line, the mean area was lower than at the walk (Table 2). Within the small and large circles, gait was not different in terms of the area (*p* = 0.09 and *p* = 0.19, respectively).

#### 3.1.2. Vertical Force

Gait (*p* = 0.007) and exercise type (*p* = 0.007) were both significant main effects for vertical force (Table 3), but leg was not (*p* = 0.71). The walk resulted in a greater mean vertical force than the trot by 14%. No two-way interactions were significant. The three-way interaction of “gait, exercise type, and leg” was significant (*p* = 0.02). At the walk and trot, there were no significant between-leg differences in the mean vertical force (*p* = 0.75 and 0.66, respectively).

At both the walk and trot, the exercise type resulted in different mean vertical force outputs, with straight-line exercise leading to a higher mean vertical force at both gaits, although only significantly higher than both circle sizes at the walk (Table 3). During straight-line exercise and small-circle exercise, the walk had a greater mean vertical force (Table 3).

#### 3.1.3. Front Hoof Pressure

Gait was a significant main effect (*p* = 0.0007, Table 4 and Table 5) for pressure, but exercise type and leg were non-significant effects (*p* = 0.16 and 0.14, respectively). The walk had a greater mean pressure than the trot by 23%. No two-way interactions were significant, but the three-way interaction of “gait, exercise type, and leg” was significant (*p* = 0.0008).

For the right and left legs, it was found that the mean walk pressure was greater than the mean trot pressure when all exercise conditions were averaged (Table 4). At the walk and trot, the exercise type did not lead to different mean pressures, but the walk had a greater mean pressure for all three exercise types compared to the trot (Table 5).

### 3.2. Dataset without Straight-Line Exercise

#### 3.2.1. Loaded Hoof Area

Gait was a significant main effect (*p* = 0.03, Table 6) for area. The canter had a 21% greater mean loaded area than the walk and a 29% greater mean loaded area than the trot. Exercise type and leg were non-significant effects (*p* = 0.78 and *p* = 0.33, respectively). The interactions of “gait and exercise type” (*p* = 0.28) as well as “exercise type and leg” (*p* = 0.72) were non-significant; however, the interaction of “gait and leg” was significant (*p* = 0.02). “Gait, exercise type, and leg” was a significant interaction (*p* = 0.0035). For the right front leg (outside leg), the canter had a greater mean loaded area than other gaits (Table 6), but this was not found in the left leg. For exercise in a large circle, the mean area loaded was different between gaits, with the canter being greater than the trot (*p* = 0.01, Table 6). While exercising in a small circle, the mean area was not different between gaits (Table 6).

#### 3.2.2. Vertical Force

Gait, exercise type, and leg were non-significant main effects (*p* = 0.33, 0.88, and 0.87, respectively) for the vertical force. “Gait and exercise type” as well as “exercise type and leg” were non-significant interactions (*p* = 0.17 and 0.71, respectively), but “gait and leg” was a significant interaction (*p* = 0.004, Table 7). “Gait, exercise type, and leg” was a significant interaction (*p* < 0.0001).

At the walk (*p* = 0.83), trot (*p* = 0.72), and canter (*p* = 0.31), the right and left legs did not have different mean vertical forces between legs. The right leg did have a lower mean vertical force at the trot than the canter (Table 7). During the large- and small-circle exercise, the mean vertical force did not differ by gait. At the walk, trot, and canter, the circle size did not have a different mean vertical force (Table 7).

#### 3.2.3. Front Hoof Pressure

Gait was a significant main effect (*p* = 0.001, Table 8) for pressure, while exercise type and leg were not (*p* = 0.59 and 0.11, respectively). The walk had 22% and 28% greater mean pressures than the trot and canter, respectively. No two-way interactions were significant. The three-way interaction of “gait, exercise type, and leg” was significant (*p* = 0.005).

For both the right and left limbs, the walk was found to have a greater mean pressure than other gaits (Table 8). In both the large and small circles, the walk was found to have a larger mean pressure than other gaits. Within each gait, there were no differences between the large and small circle sizes (Table 8).

## 4. Discussion

The objectives of this study were to determine how changes in gait and circle diameter influence area, vertical force, and pressure of the front hooves. We hypothesized that a decrease in circle diameter and an increase in speed would lead to greater differences between inside and outside limb outputs. The results determined that changes to gait more frequently lead to differences in the mean vertical force, area, and pressure outputs than changes to the circle diameter size. Most of the differences noted in this study were driven by gait, with gait being a significant effect for all evaluated outputs except for vertical force in the dataset including canter.

When evaluating gait differences, the walk typically had greater mean area and vertical force in this study, but when canter was included, the canter had the greatest mean area loaded. The walk having the greatest pressure is driven by the inclusion of the area and vertical force in the calculation of pressure. Most studies utilizing the Tekscan Hoof System for gait analysis have done so at the walk [28,29,30,32] or trot [31,33,34]. The Tekscan sensors may not be able to record data as accurately when speed increases for gaits such as the canter or even a faster trot. It is also worth exploring that adaptation to circular exercise has been previously noted as gait-specific [10]. The differences in gait may be due to increased speed or different loading patterns, such as the presence of a lead while cantering, as horses are known to protract the lead limb of a canter by flexing the elbow, carpal, hip, and tarsal joints [35]. One study found that as speed within the walk or trot increases while exercising on a treadmill, vertical impulse to the forelimbs and hindlimbs decreases [36]. While the current study did not compare speed within gaits, we did find that as gait increased from walk to trot, and therefore speed increased, the mean area, vertical force, and pressure decreased for the forelimbs. Another study found peak stress of the metacarpus and radius to be lower at a slow trot than the walk and canter and attributed the lower values of the slow trot to the symmetrical, diagonal movement of the gait [37].

Due to its two-beat diagonal footfall, the trot is considered a symmetrical gait and is the preferred gait for a lameness evaluation. The lower outputs seen at the trot in this study may be due to the fact that horses are able to utilize both forelimbs and hindlimbs within a trot stride in a more-even manner than the walk and canter [37,38,39]. The trot and canter also have moments of suspension, where the walk does not. Given that the results in this study are reported as means of the area, vertical force, and pressure, the lack of suspension in the walk could contribute to longer data collection for the right and left forelimbs at the walk than the trot and canter. Using a pressure plate, the stance phase of the walk has been found to be longer than that of a trot when tracking over both a hard surface and a soft surface [40]. Horses may also use other parts of their body, such as the musculature of the hindquarters, more so in the trot and canter than the walk, potentially leading to a decrease in the forelimb outputs [10,11,41,42]. One study found that activity of the hindlimb biceps femoris is minimal during the walk, but highly active according to electromyography at the trot and canter [43]. Another study found that at the walk and canter, horses exercising on flat and banked curves have a shorter stride length of the inside leg compared to the outside leg [10]. As horses increase in gait speed from a walk, to a trot, to a canter, it has been found that trunk muscle engagement increases as well [44].

At the trot, the mean hoof area loaded was similar regardless of exercise type, once again suggesting the trot to be the more stable gait [38,39]. Vertical force was greatest on a straight line for both the walk and trot, while pressure was not found to be different between exercise types at the walk or trot. In humans, similar results have been found where peak vertical ground reaction forces are greater in a straight line than while running around a curve [11], similar to what was found in the current study. Considering the Tekscan^TM^ sensors measure vertical forces normal (perpendicular) to the sensor, it is conceivable that shear vertical forces were higher during circular exercise. During straight-line exercise, when a horse is tracking upright, the resultant force would be vertically measured. However, when horses are tracking in a circle, as was performed in this study, lateral forces are also considered when calculating the resultant force [45]. As the sensors were worn on the front hooves and measured the force of the area that came into contact with the arena surface, only the vertical forces were included in this study. While the walk had greater vertical force in this study, other forces, such as lateral force, may be greater in the trot and canter, especially during circular exercise [45,46].

When the canter was retained in the dataset, at all gaits, the large circle did not have a different mean area, vertical force, or pressure than the small circle. When exercising in the large circle, the canter did have a greater mean area than the trot. When exercising at both small and large circles, the walk had greater pressure than both the trot and canter. With the use of a pressure plate, another study found the vertical impulse of the walk to be almost twice that of the trot on both hard and soft surfaces [40]. Comparisons between pressure plates and sensors such as the Tekscan^TM^ system should be made cautiously, as these two technologies have not been found to reliably produce the same outputs [34].

When the canter was removed from the dataset, the mean area and mean vertical force were not different between the right and left legs at the walk or trot. When the canter was maintained in the dataset, the mean loaded area of the right (outside) leg was greatest at the canter, and the mean vertical force for the right leg was greater at the canter than the trot. In this study, minimal differences were seen between limbs, but it was notable that the outside limb loaded area was greater at the canter. At racetracks with the smallest radii (>50–114 m), the outside front limb was found to have the highest number of fatal limb fractures [45]. A study evaluating body lean angle at the trot and canter lunged horses through a bitted bridle at a diameter of 10 m while wearing an inertial measurement unit on the sacrum [47]. The lean angle was reported to be greater at the canter (19°) than the trot (12°) when tracking both left and right. A greater lean-in angle at greater speeds could be cause for more push off with the outside leg [1,10], and therefore a greater mean area loaded in the outside limb at the canter, as was seen in this current study. Our findings are supported by another study, which found the third metacarpal of the outside limb endures greater strain than the inside limb when Thoroughbred horses are running around a turn [48]. While galloping around a turn, the stance phase for the inside front limb is greater, while the stance phase for the outside front limb is shorter, with larger centripetal, propulsive, and vertical forces [45]. The presence of greater peak ground reaction vertical forces in the outside leg compared to the inside leg on a curve has also been noted in humans [11]. This study did not evaluate horses tracking at a gallop, which many studies referenced in this study have evaluated. Instead, this study allowed for an exploration of gaits that are easily attainable and frequently used across the industry, including to exercise racing horses when they are not galloping. It is reasonable to expect when working a horse in a round pen or lunging a horse, especially in initial saddling and riding, that increased speed is needed to reach the optimal training state for the horse. However, given these results, the frequency of circular exercise via lunging or a round pen as a replacement to pasture turn-out or ridden exercise should be evaluated.

Circular exercise is frequently used to exercise and train animals, especially through lunging. A review of risk factors for lameness in dressage horses found lunging to be protective against lameness, while the use of walkers increased the risk of lameness [49]. Mechanical walkers are often used during recovery from lameness, so it may be difficult to separate horses that are being placed on a walker for recovery or for exercise. When on a mechanical walker, animals may be unsupervised, and are not controlled by a handler that would encourage them to travel upright and at consistent speeds. However, when lunging is utilized in dressage, often the use of a surcingle and bridle could encourage the horse to track in an upright and balanced manner, very similar to the way that a horse is “on the bit” while under saddle in dressage. In disciplines outside of dressage, lunging is typically performed with only the lunge line attached to a halter. This gives the handler less control of the horse, often resulting in lunging sessions where the horse is leaning into the circle and does not consistently engage the hindquarters and topline musculature to travel in an upright manner, making lunging in this manner less likely to be a protective factor against lameness. Circular exercise is also used under saddle for both training and competition in events such as dressage and reining. The presence of a rider is known to alter how horses utilize their back musculature at various gaits [44,50]. Further exploration into circular exercise with a rider present is needed to determine if differences between front limb outputs at the walk, trot, and canter are mitigated or exacerbated.

The current study evaluated straight line exercise versus circular exercise of a horse in a round-pen that was not attached to a lunge line. Further studies of similar design are needed to evaluate the impact of a lunge-line simply attached to a halter on forelimb disparity while an animal is in motion. When different head and neck positions were evaluated on a straight line on a treadmill, it was found that a high head position impacted limb functionality compared to an unrestrained horse [51]. Head and neck position has also been found to alter the center of motion of a horse while lunging [19]. The current study found frequent differences in gait, but limited differences in circle size for a horse moving freely in a round-pen. Differences in forelimb outputs between small and large circles may be detectable when a horse is exercised on a lunge line, as to make the circle smaller, greater tension could be applied to the lunge line attached to the horse’s halter, potentially encouraging the horse to lean in and push off more with the outside leg.

A limitation in this study is that recordings of the canter on a straight line were not attainable, and thus two sets of data were evaluated to best compare gait and exercise types. Future studies could use a long aisle-way with an appropriate surface to have horses travel in a straight line without the need of a handler. This may help to better answer the question of whether circular exercise at a canter has differences in the outside limb because of the lead or just the increase in speed. Our current study only evaluated the front limbs, which permitted us to determine differences between the inside and outside limbs. It is recognized that the hindlimbs are important in the adaptation to motion, such as turning [41,45,46], and future studies to determine the impact of gait and the circle diameter to hind limb outputs should be explored. The Tekscan^TM^ system used in the manner of this study measured vertical force and not shear force, which is certainly important for turning, especially for the hind limbs [45]. It should be noted that many technologies are utilized for gait analysis, such as vertical force plates, inertial measurement units, and sensors such as the Tekscan^TM^ Hoof System. Between studies, the technology used for analysis, as well as attachment methods and locations, is not standardized. While our study protocol and recorded metrics have been shown to be reliable [26], comparisons between studies should be made recognizing the current limitation of no standardized protocol for quantitative gait analysis for horses in motion currently.

## 5. Conclusions

While circular exercise is used frequently in the training, exercising, and competing of horses, little is known of its potential connection to joint and bone injury. This study explored the impact of gait as well as circle size to mean area, vertical force, and pressure of the front hooves. It was found that gait (walk, trot, canter) drives changes to outputs more than exercise type (straight, circular). The trot frequently had lower mean outputs than other gaits, suggesting that it is a more dynamically stable gait that could potentially allow horses to adapt to circular exercise easier than other gaits. Handlers looking to utilize circular exercise while maintaining the longevity of equine careers may consider doing so at slower gaits, as differences in outside limb output were noted at the canter, or minimizing the use of circular exercise. Future studies will help to determine if a round-pen allows the horse to adapt to changes in gait and diameter better than when exercised on a lunge line or under saddle. Lateral forces may be greater during circular exercise and should be evaluated and compared with the findings of vertical forces provided in this research.

## Figures and Tables

**Table 1 animals-11-03581-t001:** Horse details—forelimb calibration weights used for the Tekscan Hoof System and front hoof measurements used to accurately assign front hoof glue-on shoe size for each horse.

Horse	Breed	Shoe Size	Forelimb Weight (kg)	Left Hoof Width (mm)	Right Hoof Width (mm)	Left Hoof Length (mm)	Right Hoof Length (mm)
1	Stock horse	00	170	120	120	138	135
2	Stock horse	0.5	169	138	138	136	136
3	Stock horse	1.5	167	127	130	138	138
4	Stock horse	00	156	127	127	129	130
5	Stock horse	0	135	127	128	127	127
6	Arabian	1.5	139	135	130	145	140
7	Arabian	0.5	138	130	125	128	128
8	Arabian	0	131	122	135	130	125
9	Arabian	0	127	125	125	127	127

**Table 2 animals-11-03581-t002:** Mean area loaded on the front hooves for nine horses with each gait repeated three times by exercise type in the dataset excluding canter.

Exercise Type	Walk Mean Area (Sensels Loaded)	Trot Mean Area (Sensels Loaded)
Straight *	42 ^a^	35
Small circle	32 ^ab^	29
Large circle	29 ^b^	26
Within gait SEM	4	4
Within gait *p*-Value	0.01	0.11

^ab^ Means with different superscripts are significantly different within a column (*p* = 0.01). * For an exercise type, mean area was different between gaits (*p* < 0.01).

**Table 3 animals-11-03581-t003:** Mean front hoof vertical force in Newtons (N) for nine horses with each gait repeated three times by exercise type in the dataset excluding canter.

Exercise Type	Walk Mean Vertical Force (N)	Trot Mean Vertical Force (N)
Straight **	1234 ^a^	1040 ^a^
Small circle *	810 ^b^	700 ^ab^
Large circle	736 ^b^	660 ^b^
Within gait SEM	356	134
Within gait *p*-Value	0.003	0.02

^ab^ Means with different superscripts are significantly different within a column (*p* < 0.05). * For an exercise type, mean vertical force was different between gaits (*p* < 0.05). ** For an exercise type, mean vertical force was different between gaits (*p* < 0.001).

**Table 4 animals-11-03581-t004:** Mean front hoof pressure by leg in kilopascals (kPa) for nine horses with each gait repeated three times per exercise type for the dataset excluding canter.

Leg	Walk Mean Pressure (kPa)	Trot Mean Pressure (kPa)
Right leg **	723	530
Left leg **	963	775
Within gait SEM	149	149
Within gait *p*-Value	0.15	0.14

** For one leg, mean pressure was different between gaits (*p* < 0.001).

**Table 5 animals-11-03581-t005:** Mean front hoof pressure in kilopascals (kPa) for nine horses with each gait repeated three times by exercise type for the dataset excluding canter.

Exercise Type	Walk Mean Pressure (kPa)	Trot Mean Pressure (kPa)
Straight **	936	715
Small circle **	839	647
Large circle *	761	595
Within gait SEM	75	76
Within gait *p*-Value	0.09	0.31

* For an exercise type, mean pressure was different between gaits (*p* < 0.01). ** For an exercise type, mean pressure was different between gaits (*p* < 0.0001).

**Table 6 animals-11-03581-t006:** Mean area loaded on the front hooves for nine horses with each gait repeated three times by gait for the dataset excluding straight-line exercise.

Gait	Right Hoof Mean Area (Sensels)	Left Hoof Mean Area (Sensels)	Large Circle Mean Area (Sensels)	Small Circle Mean Area (Sensels)
Walk	34 ^a^	27	27 ^ab^	32
Trot	29 ^a^	26	26 ^a^	27
Canter	47 ^b^	29	39 ^b^	37
Within leg or circle size SEM	4	4	4	4
Within leg or circle size *p*-value	0.001	0.80	0.01	0.11

^ab^ Means with different superscripts are significantly different within a column (*p* < 0.05).

**Table 7 animals-11-03581-t007:** Mean front hoof vertical force in Newtons (N) for nine horses with each gait repeated three times by gait for the dataset excluding straight-line exercise.

Gait	Right Hoof Mean Vertical Force (N)	Left Hoof Mean Vertical Force (N)	Large Circle Mean Vertical Force (N)	Small Circle Mean Vertical Force (N)
Walk	745 ^ab^	804	736	813
Trot	629 ^a^	729	660	698
Canter	938 ^b^	647	817	768
Within leg or circle size SEM	94	94	84	86
Within leg or circle size *p*-value	0.01	0.27	0.20	0.40

^ab^ Means with different superscripts are significantly different within a column (*p* = 0.01).

**Table 8 animals-11-03581-t008:** Mean front hoof pressure in kilopascal (kPa) for nine horses with each gait repeated three times by gait for the dataset excluding straight-line exercise.

Gait	Right Hoof Mean Pressure (kPa)	Left Hoof Mean Pressure (kPa)	Large Circle Mean Pressure (kPa)	Small Circle Mean Pressure (kPa)
Walk	679 ^a^	924 ^a^	762 ^a^	813
Trot	484 ^b^	759 ^b^	596 ^b^	698
Canter	467 ^b^	693 ^b^	586 ^b^	768
Within leg or circle size SEM	61	61	55	86
Within leg or circle size *p*-Value	0.003	0.003	0.006	0.40

^ab^ Means with different superscripts are significantly different within a column (*p* < 0.01).

## Data Availability

Datasets used and analyzed during the present study are included in the article. Raw data and further inquiries can be directed to the corresponding author.

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
