# Peer review of "Impact of Gait and Diameter during Circular Exercise on Front Hoof Area, Vertical Force, and Pressure in Mature Horses"

_animals, 2021, doi:10.3390/ani11123581_

Round 1

Reviewer 1 Report

Dear Authors

I consider the research presented for review to be valuable to the equine community and innovative. The only problem seems to be the small group of horses on which the presented methods were tested. 

Regards

Author Response

The authors are pleased to hear that this work is valuable to the equine community and innovative as stated. We too would appreciate more animals in our study. However, in the group of horses we had access to we were restricted to a group of a horses that could trot consistently and were considered serviceably sound. There were some other horses at our university that we had access to, but unfortunately they were pacing Standardbred horses retired from racing and training careers.

Reviewer 2 Report

The paper title promises an interesting topic, the impact of gait and diameter on front hoof force and pressure in mature horses in circular exercise. This is an interesting topic and something that is a lot better understood in bipeds than in quadrupeds. The potential impact on the soundness of horses is high and in some cases is a source of risk to the horse that is reasonably easy to mitigate.

The emphasis on front hooves is reasonable since that is where most turning effort occurs, but the loading on the hind limbs is also responsible for the turning. In a gallop, and for Thoroughbred racehorses it is the left front (the lead in the turn) and the right hind limb that are most likely to be injured.

The biggest problem in this paper is the limitations to the Tekscan system. The authors fairly point out that the Tekscan measures vertical forces in a number of small areas and does not measure the shear force. This is a significant limitation to the study. A simple free-body-diagram will show that the turning force is primarily a shear force. The deformation of the surface can transfer a portion of this into a vertical load but the only forces keeping the leaning horse upright would be the centripetal force balanced by the shear horse. These diagrams exist in a number of papers including those by Dr. Hobbs and Dr. Clayton.

But wait, there is more. Taking your citation 9, from Hobbs et. al. and The Tan Et. Al. and the Peterson et. al., papers below, there is a clear pattern, surfaces and turn radius will change the balance and can shift from a traction limited to a power limited condition. If the horse is not traction limited, it is reasonable to assume that laterality of loading is minimal. But nearly the entire effect is in shear not in the vertical direction. The exception is the element. the shear loading that is converted to a vertical load due to deformation of the surface. What is most relevant is the resultant force from the shear and the vertical and the direction of that force relate to the deformed surface.

This is a worthy paper, but some thought needs to be given to how the vertical force is instructive relative to the turf radius. When using the word force it needs to be clarified that the force is perpendicular to the bottom of the hoof and that it is not the resultant. It should also be clarified that there are not three measurements, force, area, and pressure but that pressure is simply force divided by the area.

With this clarification and a discussion of the role gait plays in the measurements, the paper has value as an investigation into the vertical loads on the hoof. Some discussion of the deformation of the surface and potentially the hear force of the surface should be included. Since the composition of the surface and the maintenance conditions (see Setterbo et al) of the surface can alter it from traction limited to power limited turning which in turn changes the conditions of the experiment.

There is however value to the data. By looking at vertical forces you will see if there is resistance to overturning from the moment between the two legs. This would suggest that the most important turning effect would be shear (or traction or friction … used interchangeably and in many cases incorrectly in the literature) not vertical load. You have shown this through not having the legs be significant.

The path forward I think on this paper is to first clearly state that this is vertical loading. This should be in the title and, as long as I understand correctly, throughout the document. You need to add in free body diagrams showing the conditions of measurement and where the total load of the horse is transferred to the surface. Then look at the hypothesis and I think the strong take home is that gait impacts vertical loads but that turning does not.

If I have misread the paper sorry, but that means I will be unlikely to be the only one.

Tan H, Wilson AM. Grip, and limb force limits to turning performance in competition horses. Proc Biol Sci. 2011 Jul 22;278(1715):2105-11. doi: 10.1098/rspb.2010.2395. Epub 2010 Dec 8. PMID: 21147799; PMCID: PMC3107634.

Peterson, Michael & Sanderson, Wayne & Kussainov, Nurlan & Hobbs, Sarah & Miles, Patti & Scollay, Mary & Clayton, Hilary. (2021). Effects of Racing Surface and Turn Radius on Fatal Limb Fractures in Thoroughbred Racehorses. Sustainability. 13. 539. 10.3390/su13020539.

Author Response

The authors are appreciative of the comments and suggestions made by this reviewer. We have worked to clarify in this manuscript that force measured by the Tekscan system was only vertical force, not shear force. The title has now been reworded to “Impact of gait and diameter during circular exercise on front hoof area, vertical force, and pressure in mature horses”. Wherever “force” was previously stated in the manuscript, it has now been restated as “vertical force”

We appreciate the point this reviewer makes that while much of the turning effort occurs on the front legs, the hind legs are still involved in turning. At this point, our Tekscan system only permitted the measurement of two limbs, which we decided to utilize the front limbs as most catastrophic injury occurs to the front limbs, and we could make comparisons between the inside limb and outside limb. In future studies with this Tekscan hoof system, we, and other collaborative researchers, desire to explore these effects of gait and diameter to the hind limb outputs as well. A statement of this has been added to the limitations section of the discussion (lines 458-464).

Clarification has been made in the methods that pressure was calculated from force and area (lines 214-216).

A point has been added to the limitations of this paper that the Tekscan system will only measure vertical force (380-389), not shear force.

The suggested references have been added and included in the manuscript. We appreciate the reviewer bringing these sources to our attention (Reference 45 and 46)

In place of creating a new diagram, the excellent diagram provided in Peterson et al., 2021 has been referenced and clarification made that our sensors measured vertical force.

Reviewer 3 Report

Comments to authors

 This paper is not easy or logical to follow, there are contrasts within it, it is unnecessary long and should be more aimed.

The contrasts within the paper e.g. the conclusion and summary are not in accordance with the results. An example is concerning higher outputs in straight-line exercise type vs circular (results) while advising to use circular exercise sparely in training (conclusion, summary).

The results are in contrast with accepted knowledge of impact on limbs i.e. higher velocity (faster gaits) more impact on limbs.

Are average values of the variables measured in this study the right one to estimate impact on limbs? Or should peak values for force and pressure rather be used? Arguments for using these variables (average values instead of peak) to estimate impact on limbs lack in the paper. This is related to length of the stance phase of the gaits, walk has higher average values of the variables (pressure, force) as the limbs stay longer on the ground.

Could speed be a confounding variable of exercise type within the study / should that be mentioned or discussed more clearly ?

More comments:

Lines 122-123. Inconsistency in use of the words “gait” versus “speed”, in title vs. aim/objectives of the study.

In lines 120-123 (end of introduction/objectives): …gait not mentioned but talked about speed.

Impact of gait.. (in title) and in lines 29-31, the word gaits mentioned but not speed.

Line 153 Add type/s of scale used for weighing the horses.

Table 1. Explain the shoe size, what does it mean?

Explain also how the hoof-measurements were done, are these measurements from Tekscan program ?

Line 180, add here the length of the straight line exercise, instead of later in line 184.

Line 137, explain the abbreviation AAEP or refer to a reference explaining it.

Line 139, ad libitum should be italic font ?

Tables, suggest to decrease the number of borders between lines in all tables.

Results.

Generally could be clearer and text shorter. Do that were possible.

E.g. first line:

“Gait (P=0.03) and exercise type (P=0.03) were significant main effects for area, but leg was not (P=0.44).”

This sentence does only tell about significance but not how (walk vs trot), no numbers (results) and no information of how gait had effect on area. Numbers (results) and P-values -should be shown/related together or refer to a table with numbers.

The same applies to many more sentences in results -were P-values are given. Make results easier to follow and important to refer to tables were appropriate.

Do table 2 and 6 contain information on area? Not clear from the title of the tables but titles of columns within tables 2 and 6 contain the word area.

Lines 378-379. ”When exercising at both a small and large circle, the walk had greater pressure than both trot and canter”.

In light of this result (and more in the paper), how can it be advised in the end of summary to perform circular exercise on slower gaits?

And in Conclusion: “Handlers looking to utilize circular exercise while maintaining longevity of equine careers may consider doing so at slower gaits or minimizing the use of circular exercise”

Can this be said in light of the results (e.g. tables 2,3, and 5) of current study?

Is it not necessary to be more precise concerning the effect of each gait in summary and conclusion?

Might speed be a confounding variable of exercise type in the study?

Line 387 in discussion: ”average area loaded and average force for the right leg (outside leg) was greater at canter than other gaits”.

However, according to table 7, average force for the right leg was not different between canter and walk.

Line 389-390. “A study evaluating body lean angle at the trot and canter lunged horses through a bitted bridle at a diameter of 10 m [44]

Connect this sentence better into the text.

Lines 404-419. Interesting text/information, but lacks better connection to discussion of the results of the paper.

Lines 427-430 Discussion of results from earlier studies of effect of HNP on limbs and center of motion. HNPs of the horses in the current study have not been described, the only information given is that the horses moved freely and were not attached to a lunge line when on circles. Were the horses in current study similar in HNPs or was there great difference between individuals?

The reviewer misses more information on the horses in the current study especially on the speed (and maybe also HNP, stabilization of speed), was there great difference in the speed between individual horses, between exercise types (could speed be a confounding variable here? Faster on straight line?).

Author Response

**Author responses are included in bold**

The authors appreciate the depth of suggestions this reviewer has made to this manuscript. According with the suggestions of this reviewer and others, “force”, throughout the paper has been reworded to “vertical force” to clearly identify that the output our sensor system was able to evaluate was “vertical force”. Given this specification, our results now more clearly align with expected values from other accepted knowledge, as increased force outputs during circular exercise at faster gaits could be found in shear forces, unable to be measured in this study.

Are average values of the variables measured in this study the right one to estimate impact on limbs? Or should peak values for force and pressure rather be used? Arguments for using these variables (average values instead of peak) to estimate impact on limbs lack in the paper. This is related to length of the stance phase of the gaits, walk has higher average values of the variables (pressure, force) as the limbs stay longer on the ground. In the literature, it has been found that studies evaluating longer sessions of exercise, versus just a few steps, utilize average values instead of peak values. The authors did not feel that peak values were appropriate for this type of exercise with novel sensors, as a pinched piece of sand could report as a peak value that may not be true.

Lines 122-123. Inconsistency in use of the words “gait” versus “speed”, in title vs. aim/objectives of the study. In lines 120-123 (end of introduction/objectives): …gait not mentioned but talked about speed. Impact of gait.. (in title) and in lines 29-31, the word gaits mentioned but not speed. The word “speed” has been replaced with “gait”, as in this study speed was not evaluated, only gait (125). The word “gait” has been left in the title, as this study evaluated three gaits (walk, trot, and canter) instead of speed.

Line 153 Add type/s of scale used for weighing the horses. Scale type has been added (161)

Table 1. Explain the shoe size, what does it mean? Explain also how the hoof-measurements were done, are these measurements from Tekscan program ? Details to explain these requests have been added (150-151).

Line 180, add here the length of the straight line exercise, instead of later in line 184. Straight-line exercise length has been moved to (182-183).

Line 137, explain the abbreviation AAEP or refer to a reference explaining it. Abbreviation has been defined (139).

Line 139, ad libitum should be italic font ? Ad libitum has been changed to italic (142)

Tables, suggest to decrease the number of borders between lines in all tables. This request for fewer boarders to increase readability is appreciated, the current table formatting of this manuscript has been dictated by the journal.

Generally could be clearer and text shorter. Do that were possible.E.g. first line: “Gait (P=0.03) and exercise type (P=0.03) were significant main effects for area, but leg was not (P=0.44)”. This sentence does only tell about significance but not how (walk vs trot), no numbers (results) and no information of how gait had effect on area. Numbers (results) and P-values -should be shown/related together or refer to a table with numbers. The same applies to many more sentences in results -were P-values are given. Make results easier to follow and important to refer to tables were appropriate. Results have been shortened where possible. In the specific above example, an instruction to see Table 1 for significant results has been added (239) and similar changes have been made throughout the results.

Do table 2 and 6 contain information on area? Not clear from the title of the tables but titles of columns within tables 2 and 6 contain the word area. Titles of tables 2 and 6 now contain a specification of area

Lines 378-379. ”When exercising at both a small and large circle, the walk had greater pressure than both trot and canter”. In light of this result (and more in the paper), how can it be advised in the end of summary to perform circular exercise on slower gaits? And in Conclusion: “Handlers looking to utilize circular exercise while maintaining longevity of equine careers may consider doing so at slower gaits or minimizing the use of circular exercise” Can this be said in light of the results (e.g. tables 2,3, and 5) of current study? The walk was found to have greater vertical force (which impacted to calculation of pressure). In this study, clarification has been made that the Tekscan system is measuring vertical force, not the total force. This being said, as gaits increase to the trot and canter, other forces may have increased values that were not measured. However, in this study, differences in outputs to each limb were only seen during circular exercise, with the canter leading to greater area and force (above trot). Because of the differences in limb outputs, the authors advise that individuals perform circular exercise at gaits that do not lead to different outputs between the inside and outside limb, as different outputs between limbs could lead to abnormal loads potentially causing OA. Clarification of this point has been made throughout the discussion and the conclusion.

Is it not necessary to be more precise concerning the effect of each gait in summary and conclusion? Overall, the summary of each gait has been elaborated.

Could speed be a confounding variable of exercise type within the study / should that be mentioned or discussed more clearly ? Might speed be a confounding variable of exercise type in the study? The reviewer brings up and excellent point here. If all of the data was evaluated together, gait would be a confounding variable of exercise type in the study, as canter was only performed in circular exercise. For this reason, we used two datasets, as canter was not able to be performed safely on a straight line. One dataset contains all exercise types and only the gaits walk and trot. The second dataset contains all gaits, and only exercise types of small circle and large circle. This is certainly a limitation of this study and has been elaborated on in the discussion (454). We worked extensively with the statistical consulting center at MSU to determine that this approach to data evaluation best represented our objectives and ethically evaluated our data.

 Line 387 in discussion: ”average area loaded and average force for the right leg (outside leg) was greater at canter than other gaits”. However, according to table 7, average force for the right leg was not different between canter and walk. This statement has been reworded to clarify that canter was greater than all other gaits for area, but only the trot for force (399-401).

 Line 389-390. “A study evaluating body lean angle at the trot and canter lunged horses through a bitted bridle at a diameter of 10 m [44]. Connect this sentence better into the text. This sentence, and the sentences following have been reworded (405-407). 

Lines 404-419. Interesting text/information, but lacks better connection to discussion of the results of the paper. This section has been shortened, however, not completely removed. Circular exercise can occur in many methods, under saddle, on a lunge line, or freely in a roundpen such as performed in this study. The authors feel it is important to devote a small portion of this publication to the other methods of circular exercise, to assure that readers are aware of the different effects these types of exercise could have compared to the results found on our study. While our study is a type of circular exercise, our results are not completely parallel to the presence of a lunge line or a rider. There is little circular exercise research outside of racing, and the cited source in this paragraph (Murray, 2010) recently became popular among a group of scientists through social media as well as peaked interest within the industry.

Lines 427-430 Discussion of results from earlier studies of effect of HNP on limbs and center of motion. HNPs of the horses in the current study have not been described, the only information given is that the horses moved freely and were not attached to a lunge line when on circles. Were the horses in current study similar in HNPs or was there great difference between individuals? In studies such as these, direct comparisons between individuals are cautioned against, as each horse has a unique weight, shoe size, and hoof dimensions. 

The reviewer misses more information on the horses in the current study especially on the speed (and maybe also HNP, stabilization of speed), was there great difference in the speed between individual horses, between exercise types (could speed be a confounding variable here? Faster on straight line?). Speed was not controlled in this study, to allow horses to move freely within a gait. Other studies evaluating gait, and not speed, have also advised against controlling speed within a gait (204-208).

Reviewer 4 Report

This is a potentially valuable piece of work the results of which may inform the continued use of 'circular work' such as lunging/longeing and working a horse in a round pen. 

I have made alot of comments on the manuscript which relate to the phrasing/wording, the need for greater contextualisation throughout and some areas currently lacking in the detail required to allow the study to be replicated by independent researchers.   I hope that these are useful. 

Given the hypotheses (Ha/H1) were one-tailed the statistics need to be redone or at least significance re-determined using a p value that is less than the standard .05 that is used for two-tailed statistical analysis. 

The discussion contained useful points but needed to be reworded in places to achieve clarity.

Kind regards, I hope that the further work needed is achievable and the paper becomes publishable at some point, not least to stimulate further research in this area. 

Author Response

**Author responses are provided in bold**

This is a potentially valuable piece of work the results of which may inform the continued use of 'circular work' such as lunging/longeing and working a horse in a round pen. The authors appreciate the thorough review from this reviewer and understanding of the importance of the topic to the industry.

I have made alot of comments on the manuscript which relate to the phrasing/wording, the need for greater contextualisation throughout and some areas currently lacking in the detail required to allow the study to be replicated by independent researchers.   I hope that these are useful. The specific requested alterations have been made

Given the hypotheses (Ha/H1) were one-tailed the statistics need to be redone or at least significance re-determined using a p value that is less than the standard .05 that is used for two-tailed statistical analysis. While the authors hypothesized that certain outputs would go in a specific direction (such as decreased area for the outside limb), the results are able to be greater than or smaller than the mean. While based on literature we believed outputs such as area to be smaller for the outside limb, it does still physically have the ability to be larger, making these hypotheses a two-tailed test.

The discussion contained useful points but needed to be reworded in places to achieve clarity. Requested alterations have been made

Kind regards, I hope that the further work needed is achievable and the paper becomes publishable at some point, not least to stimulate further research in this area. 

  1. Correction has been made to “on” (17)
  2. Correction has been made to “were” (19)
  3. Clarification has been made that horses were tracking counterclockwise for the entirety of circular exercise and right leg has been renamed as” outside” leg. (23, 24, and 37)
  4. Clarification has been made that horses were tracking counterclockwise for the entirety of circular exercise and right leg has been renamed as” outside” leg. (24 and 37)
  5. Clarification has been made that horses were tracking counterclockwise for the entirety of circular exercise (23)
  6. “Lunging” has replaced “in hand” (16)
  7. Beyond assessing lameness, circular exercise is frequently used in methods such as lunging or ridden exercise. This is discussed in the introduction and discussion of this manuscript. Requested section has been reworded to “ has the potential to contribute to lameness”
  8. Reworded to “early training” (46)
  9. The “or” has not been removed, as authors wished to convey that lunging or a round pen are two separate methods to exercise in a circular manner (46).
  10. Reworded to “riding disciplines” (47)
  11. Reworded to “often” (48)
  12. Requested edits have been made (50)
  13. Requested edits have been made (51)
  14. References have been added (51)
  15. The authors were unable to read this comment
  16. Specified a mechanical walker on footing (54)
  17. Requested edits have been made (61)
  18. Requested edits have been made (74)
  19. Requested edits have been made (75)
  20. The cited study reported “peak ground reaction force” (76)
  21. Requested edits have been made (77)
  22. Requested edits have been made (78)
  23. Specification of a 5-m radius as a shortly curved track has been left in the text, as this is a smaller-than-normal radius for tracks (80)
  24. Clarification has been made, circular exercise will already be asymmetrical, but the speed and the radius can influence to what degree (82)
  25. Requested edits have been made, has been moved to second paragraph in the introduction (63-73)
  26. Placement of a semicolon at this location is not currently grammatically possible (86)
  27. Could not find a 27
  28. A discussion of sliding stops is not appropriate for this circular exercise paper, forces of the sliding stop have also not been evaluated
  29. Based on the literature found, these results show most, not all of course, Thoroughbred horses travel around a curve on their left lead when racing tracking left (95-97).
  30. “Top line” has been replaced with “back and hindlimb musculature” (106)
  31. Requested edits have been made (108)
  32. Reworded to remove assumptions (112-113)
  33. Requested edits have been made (122-123)
  34. Specified that circular exercise was in one direction (124)
  35. Horse ages have been added (133)
  36. As stated in the materials and methods, criteria for soundness was less than a grade 2 lameness as has been defined by the American Association of Equine Practitioners (139-140).
  37. The term “serviceably” has been removed (139).
  38. To best coordinate with terminology, the term “hoof sensors” was selected as this is what the Tekscan company calls the product.
  39. Same comment as above
  40. Details have been added for scale specifications (161)
  41. It is recognized that a set of pressure plates would have been better for calibrating sensors for this study, however, given our available materials, the Tekscan company felt this would be the best method to calibrate our sensors (158-162).
  42. Requested edits have been made (table 1)
  43. Shoe size has been explained (150-151 and table 1 legend)
  44. Length has been specified (183)
  45. Requested edits have been made (184-185)
  46. Requested edits have been made (189)
  47. Requested edits have been made (193-194)
  48. There were no changes in direction, all exercise was performed counterclockwise. Only Arabians and stock horses were used in this study.
  49. Requested edits have been made (214)
  50. Yes, average was “mean” (220)
  51. All left and right sensor data were exported together
  52. The authors have chosen to leave this information in, as other reviewers have requested a clear explanation for the separate datasets (226-229).
  53. Given that “was” is referring to “horse” as a single fixed effect, the singular term of “was” is appropriate in this instance (233-234).
  54. Section is labeled appropriately (237)
  55. Requested edits have been made (238)
  56. Interactions are kept in quotes for clearer understanding for a reader throughout the results.
  57. Requested edits have been made (248-249)
  58. Variance is presented as standard error of the mean
  59. Requested edits have been made (table 2 description)
  60. Requested edits have been made (256)
  61. Leg was not a significant effect for force, so averages presented in this table are combined left and right leg average (table 3)
  62. Sensels are defined as the individual cells that make up a sensor (219). This average area is reported as the amount of individual cells loaded
  63. Requested edits have been made for all superscripts
  64. Requested edits have been made (277)
  65. Correct, this section is now under the “pressure” heading and is discussing pressure results (Table 4)
  66. Did not find a comment “66”
  67. Requested edits have been made (292)
  68. Requested edits have been made (292)
  69. Requested edits have been made (289)
  70. Requested edits have been made (304)
  71. Removed “originally” (333)
  72. Correct, diameter size did not have an effect (337)
  73. Data collection of canter in a straight line was not safely possible, so no straight line canter data exists. When comparing gaits on a straight line, walk and trot are included, as these are the only gaits that performed exercise on a straight line. When comparing exercise gaits in circular exercise, walk, trot, and canter are included as all three gaits performed equal amounts of circular exercise.
  74. Connections between our results and those found in other studies are made to understand the results of this study.
  75. The reviewer brings up and excellent point here. If all of the data was evaluated together, gait would be a confounding variable of exercise type in the study, as canter was only performed in circular exercise. For this reason, we used two datasets, as canter was not able to be performed safely on a straight line. One dataset contains all exercise types and only the gaits walk and trot. The second dataset contains all gaits, and only exercise types of small circle and large circle. This is certainly a limitation of this study and has been elaborated on (453-455). We worked extensively with the statistical consulting center at MSU to determine that this approach to data evaluation best represented our objectives and ethically evaluated our data.
  76. “Topline” has been removed 368.
  77. Repetition of results has been removed
  78. Requested edits have been made (410-412)
  79. Requested edits have been made (417)
  80. Requested edits have been made (429)
  81. Requested edits have been made (440)
  82. Requested edits have been made (451)
  83. Requested edits have been made (473)
  84. The authors wish to keep the phrasing as is to stay within the results of this study

Round 2

Reviewer 2 Report

Thank you for addressing the concerns with the paper.  If you continue with this are of inquiry I hope that you find a way to characterize the dynamic loading and the shear loading on the hoof. While you emphasize the front limbs which you justify based on racehorse injuries, there is a need to close the gap between a stock horse in a round pen and a Thoroughbred race horse.  In particular a focus on symmetry would be consistent with some of the literature in human sports science since it may represent one of the most important gaps to be addressed to understand the difference between elite and the typical "club sport" test subject.  

Author Response

The authors appreciate the time this reviewer has taken to make suggestions to this manuscript. We recognize that further research into this topic is necessary, and agree with the reviewer that further exploration into the shear forces of the front limb is needed.

This study allowed us to explore vary gaits and speeds frequently used across the industry, even including racing, as horses are often worked at gaits slower than a gallop. A statement highlighting this in the discussion has been added (415-419). As the reviewer mentioned, there is certainly a gap in research outside of racing.  It is also worth noting that during early training and saddling thoroughbreds destined for race training are also exposed to circular exercise methods similar to those used in the stock horse industry to start young horses such as round pens, lunging, and mechanical walkers (44-46 and 52-53). A statement has been added to the introduction identifying Quarter Horses and Thoroughbreds as the breeds most affected by OA in the United States (114-115).

In the introduction, multiple disciplines such as dressage, reining, barrel racing, and racing are discussed. More studies are referenced on racing and dressage as these are the disciplines which the majority of studies evaluating circular exercise and running through a curve.

Reviewer 3 Report

The paper has been improved extensively.

Author Response

The authors are thankful for the detailed suggestions this reviewer provided.

Reviewer 4 Report

Thank you very much for your attention to my previous comments.  I have enjoyed reading the revised manuscript.  I just have a few final ;points that need to be addressed.   Please see the numbering on my original feedback:

The following still need to be addressed:

comment 7

comment 30

comment 43

comment 50 (and other instances average -> mean)

comment 56 (" " not always used, and in my view and experience, not needed.  Editor can advise!)

comment 57

ALSO

Lines 390 consider rewording 'maintained' as 'retained'?

Lines 450/451 - there is a word missing in this sentence.

Section heading 3.2.3  title needs expanding, pressure of what?

Author Response

The authors appreciate the time this reviewer has taken to provide suggestions for the manuscript

Comment 7: Reworded to "can be used" to show that while exercising animals in a circular manner, individual handlers can exercise their animals at varying gaits and diameter sizes, which is not standardized across all sectors of the industry (line 28)

Comment 30: "on the bit" has been removed and replaced with "travel while engaging their neck, back, and hindlimb musculature" (104-105)

Comment 43: glue-on shoes such as the product used in this study come in varying ranges of sizes. The desire for an optimal shoe fit for each horse has been added (line 153-154 and 177)

Comment 50: "average" has been replaced with "mean" throughout the manuscript

Comment 56: For the time being we would like to leave the interactions in quotes for better readability, but are happy to remove if the editor would prefer.

Comment 57: Reworded the requested statement (line 247)

389: "maintained" has been reworded to "retained"

453: This sentence has been corrected

Section heading 3.2.3. has been expanded to "front hoof pressure"